# Identification of picornavirus proteins that inhibit *de novo* nucleotide synthesis during infection

Lonneke V. Nouwen[1], Esther A. Zaal[2], Inge Buitendijk[1], Marleen Zwaagstra[1], Chiara Aloise[1], Arno L. W. van Vliet[1], Jelle G. Schipper[1], Alain van Mil[3], Celia R. Berkers[2]*, Frank J. M. van Kuppeveld[1]*

1 Section of Virology, Division of Infectious Diseases & Immunology, Department of Biomolecular Health Sciences, Faculty of Veterinary Medicine, Utrecht University, Utrecht, The Netherlands, 2 Division Cell Biology, Metabolism & Cancer, Department of Biomolecular Health Sciences, Faculty of Veterinary Medicine, Utrecht University, Utrecht, The Netherlands, 3 Experimental Cardiology Laboratory, Department of Cardiology, Regenerative Medicine Center Utrecht, University Medical Center Utrecht, Utrecht, The Netherlands

* F.J.M.vanKuppeveld@uu.nl (FJMK); C.R.Berkers@uu.nl (CRB)

## Abstract

Viruses, including picornaviruses, modulate cellular metabolism to generate sufficient building blocks for virus replication and dissemination. Previously, we showed that two picornaviruses, coxsackievirus B3 (CVB3) and EMCV, remodel nucleotide metabolism during infection. Here, we investigated whether this modulation is attributable to specific viral proteins. For this, we studied the modulation of metabolism by several recombinant CVB3 and EMCV viruses in HeLa cells. Using isotope tracing metabolomics with three distinct labels, $^{13}C_6$-glucose or $^{13}C_5/^{15}N_2$-glutamine, we reveal that the 2A protease of CVB3 and the Leader protein of EMCV inhibit *de novo* nucleotide synthesis. Furthermore, we show that nucleotide metabolism is also reprogrammed by CVB3 and EMCV in human induced pluripotent stem cell-derived cardiomyocytes. Our insights are important to increase understanding of picornavirus-host interactions and may lead to novel therapeutic strategies.

## Author summary

The picornavirus family includes well-known pathogens for humans and animals, such as enteroviruses (e.g., poliovirus, coxsackievirus, rhinovirus), cardioviruses (e.g., encephalomyocarditis virus [EMCV] and Saffold virus) and aphthoviruses (e.g., foot-and-mouth disease virus [FMDV]). In humans, picornaviruses and especially enteroviruses can cause a variety of diseases, ranging from hand-foot-and-mouth disease, myocarditis, and conjunctivitis to aseptic meningitis and acute flaccid paralysis. In animals, EMCV causes encephalitis, myocarditis and FMDV causes foot-and-mouth disease, which have great impact on animal health as well as on the global life stock industry. Upon host infection, picornaviruses

**Data availability statement:** All relevant data are within the manuscript and its Supporting Information files.

**Funding:** ◦ Work in the labs of FJMvK and CRB on viral modulation of cellular metabolism is supported by funds (awarded to FJMvK) of CARE, an EC project that has received funding from the Innovative Medicines Initiative 2 Joint Undertaking (JU) under grant agreement n°101005077. FJMvK received funding from European Union's Horizon 2024 ERC Advanced Grant program under grant agreement No 01053576. AvM is supported by the EU-funded project BRAV3 (H2020, ID:874827). The funders had no role in study design, data collection and analysis, decision to publish, or preparation of the manuscript.

**Competing interests:** The authors have declared that no competing interests exist.

manipulate various cellular processes to optimize replication and dissemination, including the modulation of host metabolism. We have previously shown that nucleotide metabolism is modulated during CVB3 and EMCV infection in HeLa cells. Our current study reveals that also in in human induced pluripotent stem cell-derived cardiomyocytes nucleotide metabolism is modulated and that two picornavirus proteins, the 2A protease of CVB3 and the Leader protein of EMCV, actively inhibit *de novo* nucleotide synthesis. Elucidating picornaviral modulation of cellular metabolism is important for our comprehension of virus-host interactions and may unveil novel therapeutic targets.

## Introduction

The family *Picornaviridae*, a large and diverse group of small, non-enveloped, positive-sense RNA viruses, contains many pathogens that affect to human and animal health [1–3]. Among others, this family includes the genus *Enterovirus*, the best-known of which is poliovirus, the cause of poliomyelitis. In addition, coxsackie A and B viruses (e.g., CVB3), echoviruses, several numbered enteroviruses, including emerging viruses such as EV-D68 and EV-A71 that pose serious global health problems, as well as rhinoviruses belong to this genus. Collectively, these viruses cause diseases ranging from acute flaccid meningitis, encephalitis, hand-foot and mouth disease to common colds [4–6]. Other genera in the picornavirus family are the genus *Cardiovirus* or *Aphthovirus* that include viruses such as encephalomyocarditis virus (EMCV), Theiler's virus (TMEV), Saffold virus (SAFV), and foot-and-mouth disease virus (FMDV), of which the latter has a tremendous impact on animal health as well as the global livestock industry [7,8].

Following entry of picornaviruses in host cells, the single-stranded positive-sense RNA genome (6.7-10.1 kb) is released into the host cell cytoplasm [9]. The genome is directly translated, yielding a polyprotein that is autocatalytically cleaved by one (3C$^{pro}$) or two viral proteases (3C$^{pro}$ and 2A$^{pro}$). This yields both the structural (VP1–4) and non-structural viral proteins (2A-C and 3A-D) essential for various stages of the viral life cycle. Several picornaviruses encode for an additional protein, L or L$^{pro}$, upstream of the structural proteins. Picornaviruses actively modulate cellular processes to ensure efficient replication and spreading. For enteroviruses, 2A$^{pro}$ is an important modulator of these cellular processes. 2A$^{pro}$ has various and essential roles in viral replication and host-cell interactions [10,11]. Early in infection, 2A$^{pro}$ induces the cleavage of eukaryotic initiation factor 4G I (eIF4GI) and NUP98, inducing host-translational shut-off and nucleocytoplasmic trafficking disorder (NCTD) [12]. Apart from these effects, 2A$^{pro}$ plays an important role in evading cellular antiviral responses, which contributes to efficient infection and replication [10,11]. The cardioviruses and aphthoviruses also encode proteins that counteract antiviral responses during infection, namely L and L$^{pro}$, respectively. Although these proteins share several functionalities, there are also differences. L causes NCTD, but does not induce cleavage of eIF4GI, while L$^{pro}$ induces host-translational shut-off by cleaving eIF4GI,

but does not induce NCTD [13–16]. Apart from its more conventional roles, 2A$^{pro}$ has also been reported to be involved in the import of fatty acids, suggesting that 2A$^{pro}$ has a role in regulating cellular metabolism [17].

It is increasingly appreciated that cellular metabolism is modulated during viral infection. This modulation of host metabolism is thought to sustain virus replication and spreading and to be involved in the evasion of innate immune pathways. Which metabolic pathways are affected during infection, and how and for what reason these pathway are affected differ per virus, but in general viruses affect key metabolic pathways, including the metabolism of glucose, glutamine, nucleotides and lipids [18–28]. Moreover, several studies have shown that also picornaviruses modulate these metabolic pathways during infection [29–34]. In our previous work, we have used a metabolomics approach to study the alterations in cellular metabolism upon picornavirus infection and provided evidence that nucleotide metabolism is affected during infection [34].

We here extend this exploration by questioning whether specific viral proteins could be responsible for the alterations in nucleotide metabolism during infection. Since 2A$^{pro}$ of enteroviruses changes many cellular processes during infection, we wondered whether 2A$^{pro}$ could have any effect on nucleotide metabolism. To study this, isotope tracing experiments with $^{13}C_6$-glucose or $^{13}C_5$/$^{15}N_2$-glutamine were performed in HeLa R19 cells and in human induced pluripotent stem cell-derived cardiomyocytes (hiPSC-CMs) infected with both wild type CVB3 (CVB3) and a recombinantly generated CVB3 virus with a catalytically dead 2A protein (CVB3–2Am). We demonstrate that CVB3 reprograms nucleotide metabolism, both in HeLa R19 cells and in hiPSC-CMs, and that 2A$^{pro}$ is involved in restricting *de novo* nucleotide synthesis. With additional $^{13}C_6$-glucose isotope tracing studies, we provide evidence that L also restricts *de novo* nucleotide synthesis, while L$^{pro}$ does not. A better understanding of the connections between picornaviruses and host metabolism provides insights into picornavirus-host interactions which can lead to novel therapeutic strategies.

## Results

### CVB3 and CVB3–2Am infection increases the levels of purine and pyrimidine metabolites in HeLa cells

2A$^{pro}$ of CVB3 regulates a myriad of processes during infection. Since 2A$^{pro}$ has also been described to affect fatty acid metabolism, we wondered whether 2A$^{pro}$ could affect other metabolic pathways and more specifically, nucleotide metabolism [17]. To investigate this, we performed an isotope tracing study using $^{13}C_6$-glucose in HeLa R19 cells infected with both wild type CVB3 (CVB3) and a recombinantly generated CVB3 virus with a catalytically dead 2A protein (CVB3–2Am). A comparison of total metabolite levels (i.e., the sum of all labeled and unlabeled isotopologues per metabolite) between CVB3/CVB3–2Am and mock-infected cells showed that infection with both CVB3 and CVB3–2Am increases the levels of nucleobases, nucleosides and nucleotide monophosphates, albeit that this effect is delayed with approximately two hours during CVB3–2Am infection (Fig 1). A similar delay has been observed before during our proteomic studies and reflects a delay in replication of CVB3–2Am compared to CVB3 [35]. Moreover, levels of dihydroorotate and N-carbamoyl-aspartate decreased, both of which are early intermediates in the *de novo* pyrimidine synthesis pathway. These data are in line with our previous results, which indicated that CVB3 increases the levels of purine and pyrimidine metabolites during infection and that this increase is the result of increased nucleotide degradation and salvage rather than of *de novo* nucleotide synthesis. Thus, CVB3 and CVB3–2Am infections alter host metabolism in a highly similar fashion, with the most pronounced alterations observed in nucleotide metabolism.

### CVB3 infection limits *de novo* nucleotide synthesis, while CVB3–2Am does not

We next analyzed the incorporation of $^{13}$C-carbon from glucose into nucleotide mono-, di-, and triphosphates. Nucleotides can be synthesized through *de novo* nucleotide synthesis or nucleotide salvage pathways, and also degradation of DNA or RNA can contribute to overall nucleotide pools. The labeling patterns in nucleotide synthesis intermediates differ depending on the pathways involved, allowing determination of activated and inhibited pathways. During *de novo* synthesis, $^{13}$C-glucose labels the ribose moiety through the pentose phosphate pathway (PPP), which is a relatively

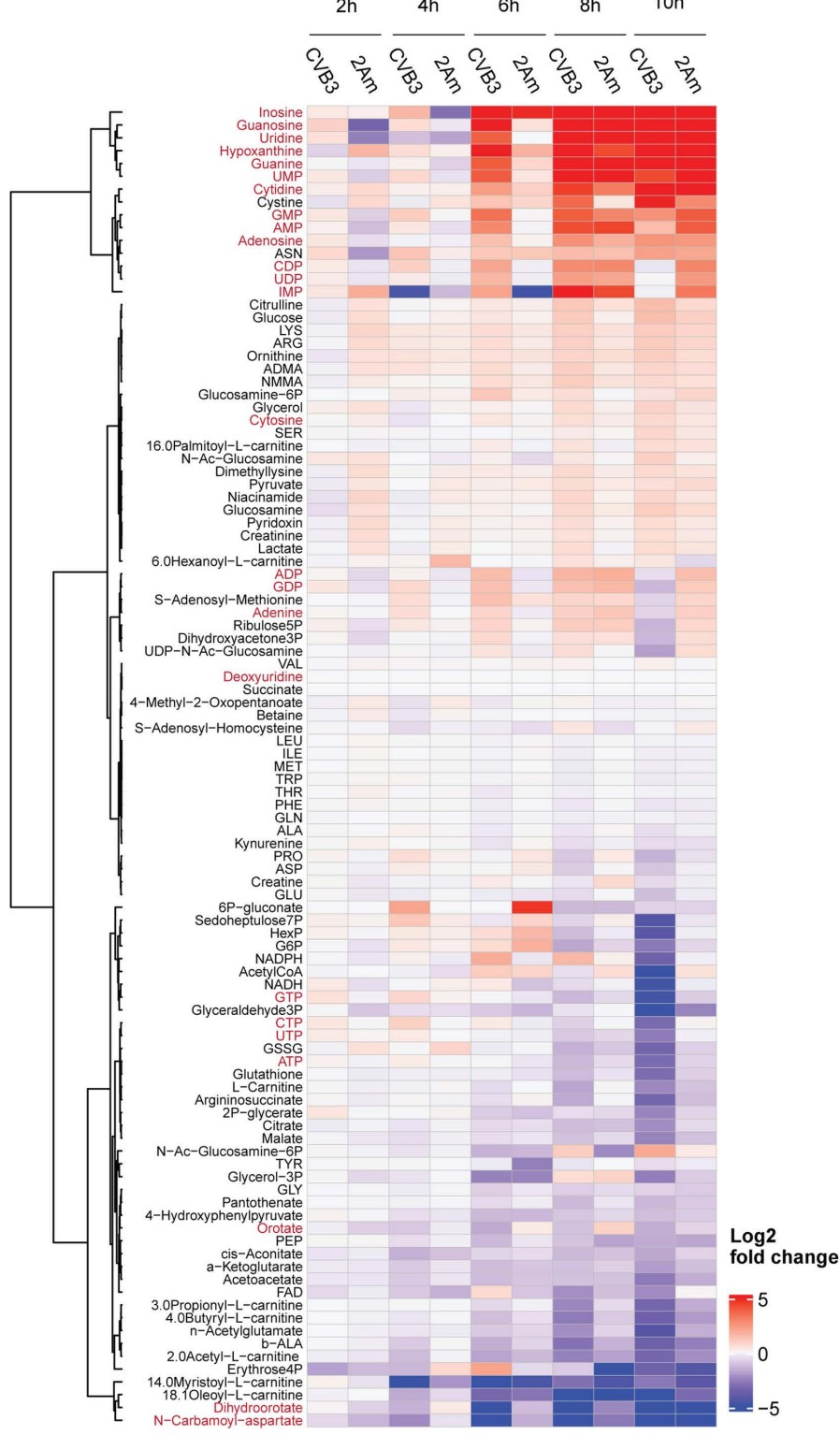

**Fig 1. CVB3 and CVB3-2Am infection increase the levels of purine and pyrimidine metabolites in HeLa cells.** 13C-glucose isotope tracing study in mock-, CVB3- and CVB3-2Am infected HeLa R19 cells (MOI 5, three replicates). Cells were infected, lysed at 2, 4, 6, 8, or 10 hours post infection (hpi) and measured by LC-MS to identify metabolites and quantify the different isotopologues. Heatmap of log2 fold changes of the total levels (i.e., the

sum of all isotopologues) of the indicated metabolites between CVB3- or CVB3-2Am- and mock-infected cells. Log2 fold changes are calculated based on the mean of three replicates. Red metabolites are metabolites in the purine and pyrimidine pathway.

fast process. In addition, the nucleobase is labeled by $^{13}C$-glucose, resulting in nucleotides with either 2–4 carbons (when only the nucleobase is labeled) or >5 labeled carbons (when both ribose and nucleobase are labelled). When the salvage pathway is active, nucleobases can undergo recycling to form nucleosides and nucleotides by coupling with a (labeled) ribose, yielding nucleotides with 5 labeled carbon atoms, but not exceeding 5. Unlabeled nucleotides either preexisted in the cell before labeling commenced or were released through the degradation of nucleic acids and nucleotides (Fig 2A).

Comparing mock versus CVB3 infection over time, we observed a decrease in the M > 5 fractions and an increase in the M + 0 fraction during infection, as exemplified by UDP labeling patterns (Fig 2B, left panel), accompanied with an increase in the total levels of UDP (Fig 2B, right panel). These patterns are indicative of a decrease in *de novo* nucleotide synthesis with a concomitant increase of nucleic acid degradation, in line with our previous work [34]. In contrast, comparing CVB3 with CVB3–2Am infection revealed a distinct labeling pattern induced by CVB3 compared to CVB3-2A, with labeling patterns in CVB3-2A resembling those in mock-infected cells. Compared to CVB3 WT, an increase in the levels of both M > 5 and M + 5 isotopologues was observed in CVB3-2A-infected cells and a decrease in the relative contribution of the M + 0 isotopologue to the total pool. However, like in CVB3 infected cells, the absolute levels of M + 0 isotopologues increased in CVB3–2Am-infected cells, albeit with a delay (Fig 2B). Labeling patterns in other purine and pyrimidine metabolites upon infection with CVB3 WT and -2A mirrored those found in UDP (Fig 2C). These data indicate that although both CVB3 and CVB3–2Am increase purine and pyrimidine metabolite levels during infection, there is a restriction of *de novo* nucleotide synthesis during CVB3, but not during CVB3–2Am infection, suggesting that 2A$^{pro}$ is involved in restricting *de novo* nucleotide synthesis.

### Glutamine tracing studies confirm a role of 2A$^{pro}$ in restricting *de novo* nucleotide synthesis

While $^{13}C_6$-glucose isotope tracing studies can explore the potential preferences within nucleotide metabolic pathways, the contribution of salvage and *de novo* nucleotide synthesis pathways cannot be completely deconvoluted, as both result in the formation of M + 5 isotopologues. To validate the proposition that 2A$^{pro}$ restricts *de novo* nucleotide synthesis, we next performed $^{13}C_5$-glutamine and $^{15}N_2$-glutamine tracing studies, as glutamine exclusively labels the nucleobases of nucleotides. $^{13}C_5$-glutamine labels pyrimidine nucleobases only and can give rise to M + 1–3 nucleotides, while $^{15}N_2$-glutamine labels both purine and pyrimidine nucleobases, leading to nucleotides with an M + 2–5 or M + 1–3, respectively (Fig 3A) [37]. Two of the five nitrogen atoms in purine nucleobases will be labeled by $^{15}N_2$ glutamine via aspartate or glycine. As glycine is barely labeled by $^{15}N_2$-glutamine due to the availability of the precursor serine in the medium (S1 Fig), we assume labeling of purine nucleobases via glycine will not occur. Therefore, in our experimental setup, $^{15}N_2$-glutamine labeling will lead to purine nucleotides with an M + 2–4 (Fig 3A).

With all three labels, the total labeled fraction of ADP and UDP in CVB3-infected compared to CVB3–2Am-and mock-infected cells was decreased (Fig 3B). This confirms that in contrast to CVB3, CVB3–2Am does not restrict *de novo* nucleotide synthesis. In addition, both the levels and labeling of pyrimidine synthesis intermediates N-carbamoyl-aspartate and orotate differed between CVB3 and CVB3–2Am infected cells (Fig 3C). In both CVB3 and CVB3–2Am infected cells, the N-carbamoyl-aspartate levels decreased. In contrast, orotate levels decreased during CVB3, but not CVB3–2Am infection. The total labeled fractions of both N-carbamoyl-aspartate and orotate were lower in CVB3- compared to CVB3–2Am infected cells (Fig 3C). This suggests that while production of N-carbamoyl-aspartate and orotate is restricted in CVB3 infected cells, production of these metabolites is increased in CVB3–2Am infected cells, further confirming that 2A$^{pro}$ is involved in restricting *de novo* nucleotide synthesis.

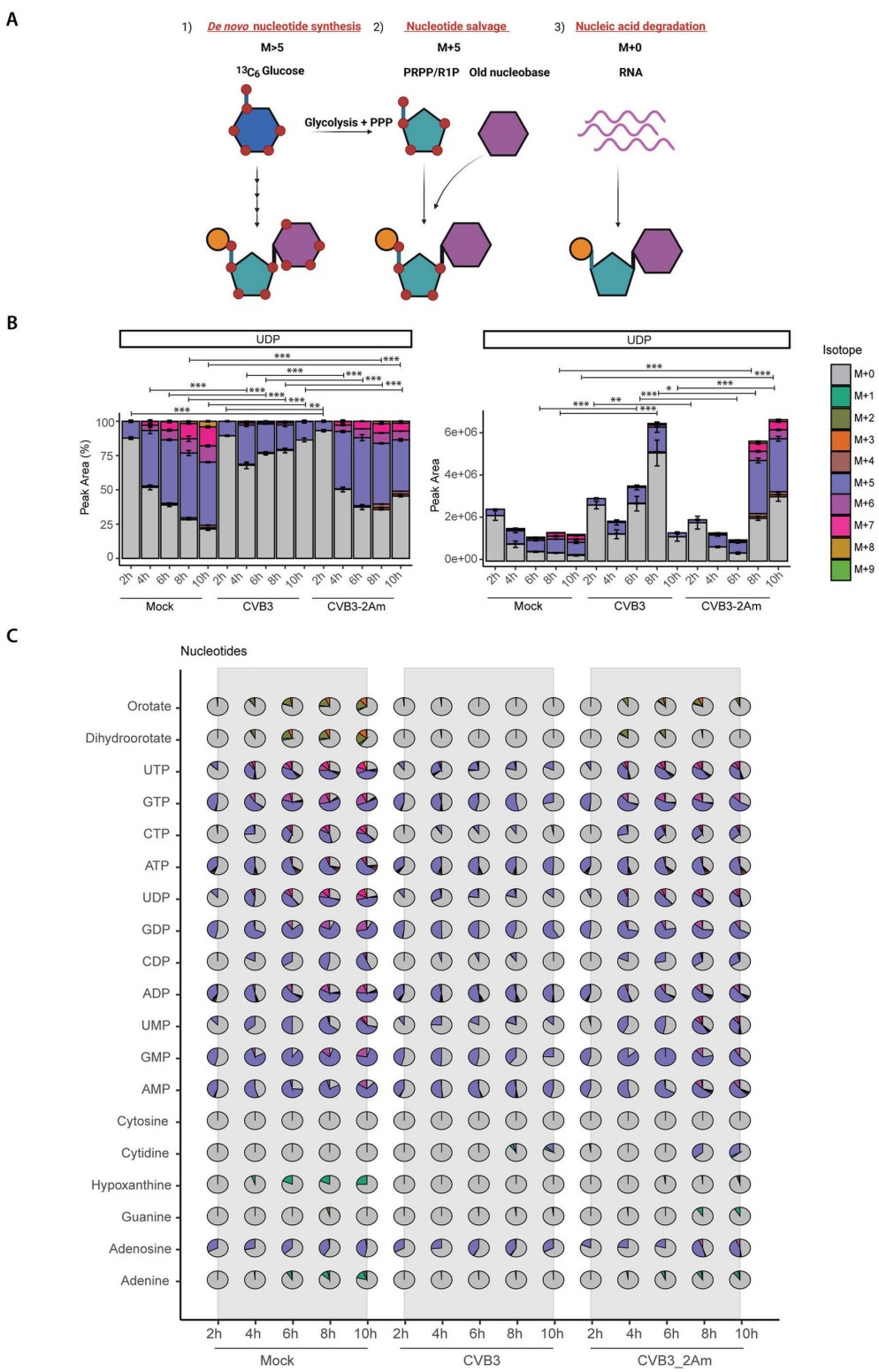

**Fig 2. CVB3 infection limits *de novo* nucleotide synthesis, while CVB3-2Am does not.** 13C-glucose tracing study in mock-, CVB3- and CVB3-2Am infected HeLa R19 cells (MOI 5, three replicates). Cells were infected, lysed at 2, 4, 6, 8, or 10 hpi and measured by LC-MS to identify metabolites and

quantify the different isotopologues. A) Schematic representation of nucleotide labeling by $^{13}C_6$-glucose [36]. Nucleotides can be synthesized de novo, released during degradation of nucleic acids, or recycled (i.e., called the salvage pathway), resulting in the formation of different isotopologues. PRPP, phosphoribosyl pyrophosphate; R1P, ribose-1-phosphate. Orange = phosphate group; Turquoise = pentose sugar; Purple = nucleobase. B) Isotopologue distribution and absolute peak areas of UDP. For statistical analysis, linear mixed effect models with an interaction of time and treatment and a random effect of replicate were performed. p-values between specific groups were calculated by performing a contrast analysis, in which either the total labeled fraction or the total peak area between groups was compared. *$p < 0.05$, **$p < 0.01$, ***$p < 0.001$. C) Isotopologue distribution of detected purine and pyrimidine metabolites.

## Glutamine tracing studies reveal an activation of nucleotide salvage in CVB3 infected cells

Interestingly, our metabolomic analysis with $^{15}N_2$-glutamine in CVB3- and CVB3–2Am infected cells revealed an increase in the levels of M + 1 fraction of purines and especially guanosine monophosphate (GMP) during CVB3 and CVB3–2Am infection (Fig 3D, dark green bars), which is unlikely to be derived from de novo nucleotide synthesis. During de novo nucleotide synthesis, four nitrogen atoms can be derived from glutamine, directly or via aspartate. Direct labeling from glutamine (2 nitrogen atoms) likely occurs from the start of the infection, whereas aspartate, which donates 2 additional nitrogen atoms, first needs to be synthesized/labeled by glutamine, which requires time. Consequently, one would anticipate that de novo purine synthesis results in at least two labeled nitrogen atoms, increasing over time to three or four when aspartate also gets labeled. The M + 1 fraction likely arises from the activation of nucleotide salvage. During the recycling of unlabeled hypoxanthine to inosine monophosphate (IMP), IMP can be utilized to form either adenosine monophosphate (AMP) or GMP. In this process, one nitrogen atom is provided by glutamine (in case of GMP) or aspartate (in case of AMP)(Fig 3E). Thus, the observed increase in the M + 1 fraction of purines and particularly in GMP strongly suggests that nucleotide salvage is activated during CVB3 infection and, to a lesser extent, during CVB3–2Am infection.

## 2A$^{pro}$ and L restrict de novo nucleotide synthesis during infection, while L$^{pro}$ does not

Because we observed that 2A$^{pro}$ limits de novo nucleotide synthesis we wondered whether proteins with similar functionalities would also affect the de novo nucleotide synthesis. As previously described, 2A$^{pro}$ of enteroviruses, L of cardioviruses and L$^{pro}$ of aphthoviruses are involved in inhibiting host stress- and anti-viral responses. These proteins suppress host anti-viral responses via different mechanisms and exhibit several different functionalities. 2A$^{pro}$ causes both NCTD and host-translational shut-off by cleaving eIF4GI, L causes NCTD, but does not cleave eIF4GI and L$^{pro}$ does not cause NCTD, but does cleave eIF4GI (Fig 4A). To dissect whether either NCTD or eIF4G cleavage could cause inhibition of de novo nucleotide metabolism during infection, we made use of recombinantly generated CVB3–2Am + L and CVB3–2Am + L$^{pro}$, which lack the catalytic activity of 2A$^{pro}$, but include either a functional L from EMCV or a functional L$^{pro}$ of FMDV (Fig 4B). As one might argue that the increase in de novo nucleotide synthesis during CVB3–2Am infection compared to CVB3 infection can be (partially) explained by the growth delay that CVB3–2Am exhibits compared to CVB3, an additional mutant, CVB3–2Bm, was added to the experiment. CVB3–2Bm contains a double lysine mutation specifically in the 2B protein (K[41, 44]L) of CVB3, but does contain a functional 2A$^{pro}$. Mutating the 2B protein delays CVB3 replication and virus production [40]. Therefore, CVB3–2Bm can thus be used to exclude growth defects underly the observed phenotype (S2 Fig). To track nucleotide synthesis pathways during infection of HeLa R19 cells with CVB3, CVB3–2Am, CVB3–2Am + L, CVB3–2Am + L$^{pro}$ and CVB3–2Bm a $^{13}C_6$-glucose tracing study was performed.

During CVB3 and CVB3–2Bm infection, but not during CVB3–2Am infection, a decrease of the M + 5 and M > 5 fractions was observed. This suggests that the inhibition of de novo nucleotide synthesis is not attributable to delayed growth but linked to the proteolytic activity of 2A$^{pro}$. Interestingly, isotopologue labeling patterns of UTP and GTP during CVB3–2Am + L infection mirrored those in CVB3 WT-infected cells, with a relative decrease of the M + 5 and M > 5 fractions compared to mock-infected cells. In contrast, CVB3–2Am + L$^{pro}$ labelling patterns resembled those in mock-infected cells (Figs 4C, 4D and S3A). This suggests that L, but not L$^{pro}$, restricts de novo nucleotide synthesis. Moreover, while the levels of

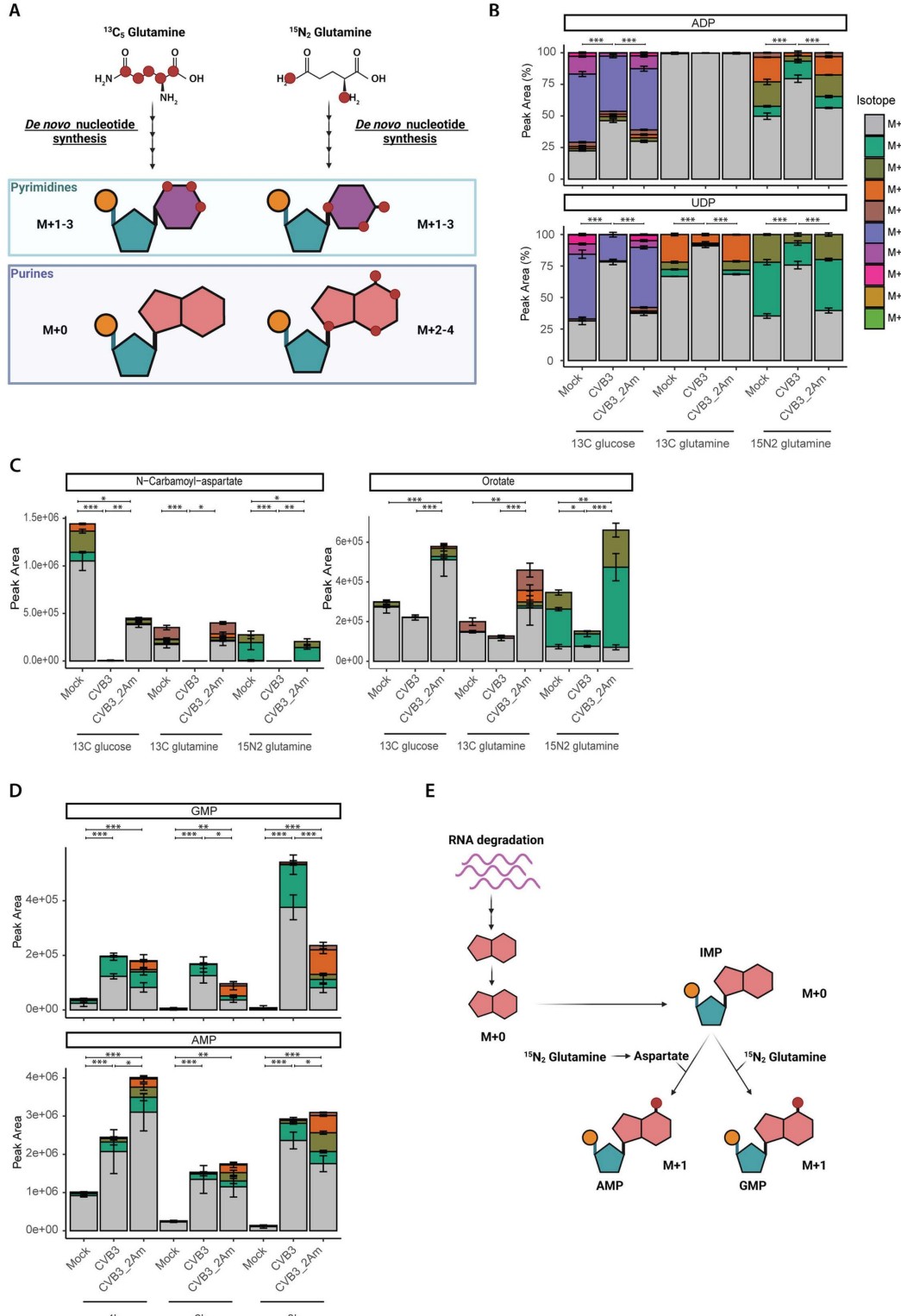

**Fig 3. Glutamine tracing studies confirm a role for 2A^pro in restricting the de novo nucleotide synthesis.** $^{13}C_6$-glucose, $^{13}C_5$-glutamine and $^{15}N_2$-glutamine isotope tracing study in mock-, CVB3-2Am- and CVB3-infected HeLa R19 cells (MOI 5, three replicates per treatment and per iso-tope). Cells were infected, lysed at 4, 6 or 8 hpi and measured by LC-MS to identify metabolites and quantify the different isotopologues. A) Schematic

representation of nucleotide labeling by $^{13}C_5$-glutamine and $^{15}N_2$-glutamine [38]. B) Isotopologue distribution of ADP and UDP at 6 hpi. C) Peak areas of N-carbamoyl-aspartate and orotate at 6 hpi. D) Peak areas of GMP and AMP labeled with $^{15}N_2$-glutamine. E) Schematic of hypoxanthine released from RNA degradation that can be salvaged to IMP and then be converted to AMP and GMP [39]. During this latter conversion, one (labeled) nitrogen atom is added. For statistical analysis, linear mixed effect models with an interaction of time and treatment and a random effect of replicate were performed. For N-carbamoyl-aspartate, a normal distribution of the residuals could not be assumed and therefore a non-parametric linear mixed effect model with an interaction of time and treatment and a random effect of replicate was performed. p-values between specific groups were calculated by performing a contrast analysis, in which total labeling, being either total labeled fraction (B) or total peak area (C), or peak areas of the M + 01 isotopologue (D) were compared between groups. *$p < 0.05$, **$p < 0.01$, ***$p < 0.001$.

the pyrimidine synthesis intermediate N-carbamoyl-aspartate decrease during infection with all viruses, this effect is faster and more pronounced in CVB3, CVB3-2B and CVB3–2Am + L as compared to CVB3–2Am and CVB3–2Am + L$^{pro}$ (Figs 4D and S3B).

To verify these findings, we used an EMCV recombinant system. EMCV-L$^{zn}$ is a mutant virus that has mutations in the zinc finger domain of L, thereby inactivating L protein functionalities and providing a tool to study 2A$^{pro}$ and L$^{pro}$ in an infection setting [11,41,42]. A $^{13}C_6$-glucose tracing study in cells infected with EMCV, EMCV-L$^{zn}$, EMCV-L$^{zn}$ + 2A$^{pro}$ and EMCV-L$^{zn}$ + L$^{pro}$ was performed. The data of the mock and EMCV infected cells have been presented in our earlier work [34]. We here show this data again to allow for a comparison of the results with the mutant viruses to mock- and EMCV-infected cells. Isotope tracing revealed a decrease in the M + 5 and M > 5 fractions of ATP and UTP during EMCV and EMCV-L$^{zn}$ + 2A$^{pro}$ infection, but not during infection of EMCV-L$^{zn}$ and EMCV-L$^{zn}$ + L$^{pro}$ infection (S4C and S4D Fig). Similarly, the levels of N-carbamoyl-aspartate decreased during infection with all viruses and this decrease occurred faster during infection with EMCV and EMCV-L$_{zn}$ + 2A$^{pro}$ (S4E Fig). The levels of orotate also decrease with the strongest decrease in EMCV and EMCV-L$_{zn}$ + 2A$^{pro}$ infected cells (S4E Fig). Together, these data suggest that both 2A$^{pro}$ and L have the capacity to restrict *de novo* nucleotide synthesis, while L$^{pro}$ does not possess this functionality.

## CVB3 and EMCV-induced metabolic alterations in hiPSC-CMs mirror those in HeLa cells

HeLa cells are derived from cancerous tissue characterized by an aberrant metabolic profile [44–47]. Consequently, it is plausible that specific metabolic alterations induced by picornaviruses may be masked or differ compared to primary cells. In pursuit of a more physiologically relevant context for studying picornavirus metabolism and because CVB3 and EMCV are known to be able to infect cardiomyocytes and cause myocarditis, we opted for human induced pluripotent stem cell-derived cardiomyocytes (hiPSC-CMs) [7,48]. Characterization of hiPSC-CMs using flow cytometry and immunofluorescence staining for Troponin T shows high purity of cardiomyocytes (>96.5%) (S5A and S5B Fig). Importantly, hiPSC-CMs were infected efficiently by both CVB3 and EMCV (S5C and S5D Fig). We therefore performed glucose tracing studies to dissect the metabolic alterations induced by CVB3, CVB3–2Am and EMCV infection in hiPSC-CMs, comparing CVB3 to EMCV or CVB3–2Am infected cells in two independent experiments.

In general, we noticed more variation between experiments in hiPSC-CMs compared to the experiments done with HeLa R19 cells; a principal component analysis (PCA) on both cell types revealed that whereas HeLa R19 samples cluster based on treatment, hiPSC-CMs samples cluster based on experiment (S6A and S6B Fig). Despite this variation, we could observe important similarities between HeLa R19 and hiPSC-CMs when comparing total metabolite levels. An increase in the levels of purine and pyrimidine metabolites was observed in hiPSC-CMs infected with either CVB3 or EMCV (Figs 5A and S7A), albeit at different time points. Concurrently with this increase, a decrease in N-carbamoyl-aspartate levels was observed (Figs 5A and S7B). Moreover, in the CVB3–2Am infected cells, the effect on purine and pyrimidine metabolite levels was lost (Fig 5A). The effects on *de novo* nucleotide metabolism were smaller in hiPSC-CMs compared to HeLa R19 cells, based on isotopologue labeling patterns. The total labelled fraction of nucleotides decreased slightly during CVB3 and EMCV infection (Figs 5B and S7C), an effect that was lost during CVB3–2Am infection. Notably, hiPSC-CMs exhibit generally lower rates of *de novo* nucleotide synthesis compared to HeLa R19 cells, which is not

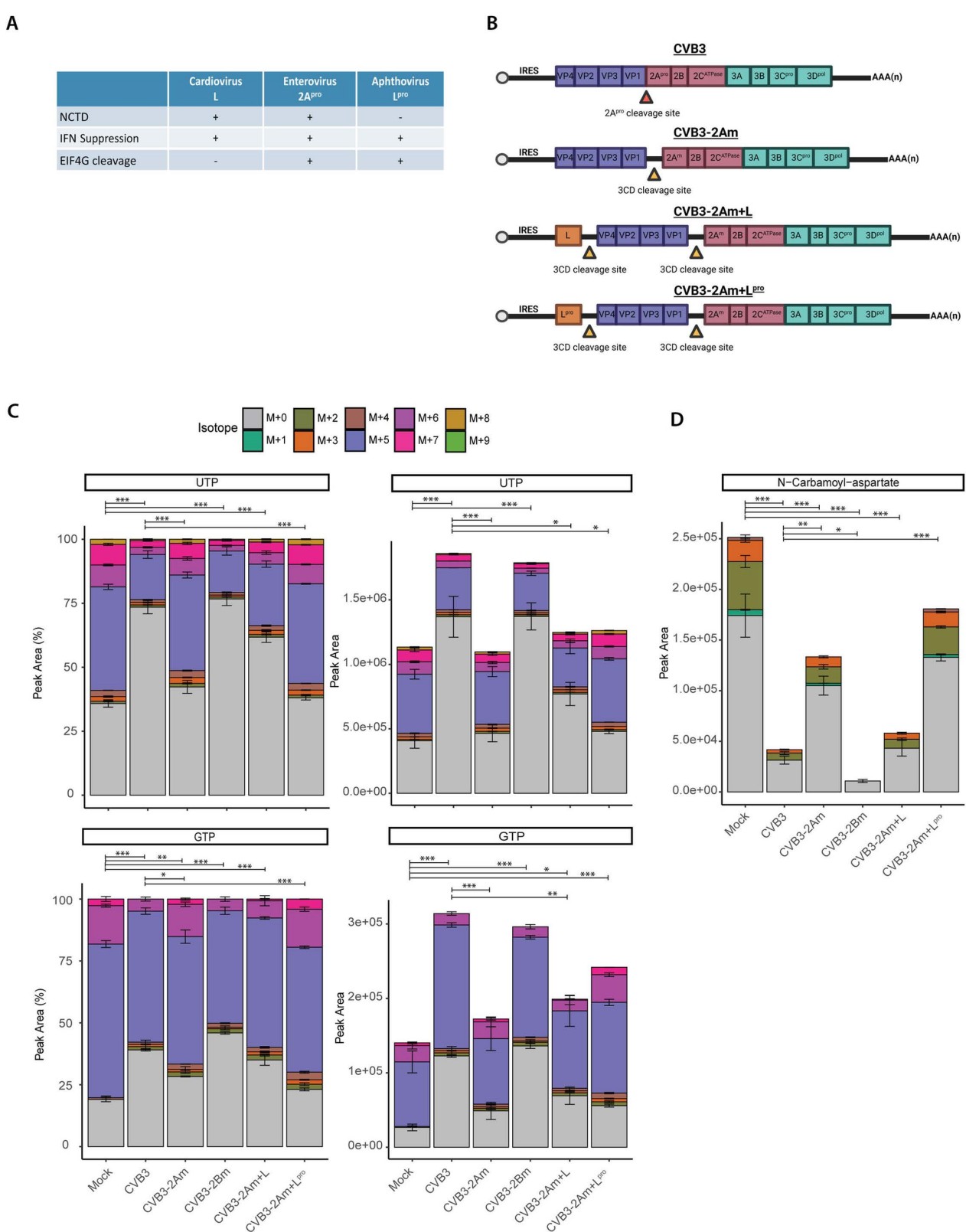

**Fig 4. 2A^pro and L restrict de novo nucleotide synthesis, while L^pro does not.** $^{13}C_6$-glucose isotope tracing study in mock- and CVB3-, CVB3–2Am, CVB3–2Am+L, CVB3–2Am+L^pro infected HeLa R19 cells (MOI 5, three replicates). Cells were infected, lysed at 6 or 8 hpi and measured by LC-MS to

identify metabolites and quantify the different isotopologues. Data obtained at 8 hpi are shown here. A) table of L, 2Apro and Lpro functions. B) Schematic representation of CVB3 and the CVB3 recombinant viruses that were used in this study [43]. Fig 4B was created with BioRender. C) Peak areas and isotopologue distribution of UTP and GTP at 8 hpi. D) Peak areas of N-carbamoyl-aspartate at 8 hpi. For statistical analysis, linear mixed effect models with an interaction of time and treatment and a random effect of replicate were performed. A rank transformation on the data was performed to ensure a normal distribution of the residuals. For GTP, a normal distribution of the residuals could not be assumed for the fractional data and therefore a non-parametric linear mixed effect model with an interaction of time and treatment and a random effect of replicate was performed. p-values between specific groups were calculated by performing a contrast analysis, in which total labeling (being either total labeled fraction or total peak area) was compared between groups. *$p < 0.05$, **$p < 0.01$, ***$p < 0.001$.

surprising considering the non-dividing nature of these hiPSC-CMs (Fig 5B). The absence of an active *de novo* nucleotide synthesis pathway likely masked an effect of viral infection on this pathway. Together, these results indicate that the changes of purine and pyrimidine metabolite levels induced by CVB3 and EMCV in hiPSC-CMs resemble those in HeLa R19 cells, albeit with a higher degree of variation.

To reveal if other pathways apart from nucleotide metabolism changed in hiPSC-CMs, we performed a pathways impact analysis, comparing the impact of CVB3 infection with mock infection cells across both experiments (Fig 6A). Indeed, CVB3 infection resulted in a clear metabolic phenotype across experiments, in which primarily amino acid pathways were altered, including arginine, proline and alanine, and aspartate and glutamate pathways (Fig 6A). We therefore looked into the virus-induced changes at the level of individual amino acids related to these pathways and clustered amino acids in clusters based on their relative changes during the course of CVB3 infection. Levels of amino acids in clusters 1 and 3 (alanine, aspartate, glutamate, glycine, serine, glutamine) either decreased or were stable during CVB3, CVB3–2Am and EMCV infection, while levels of amino acids in clusters 2 and 4 (arginine, tryptophan and threonine) were consistently higher (Figs 6B, 6C and S8). Interestingly, proline seems to increase early in infection and to decrease later in infection. Notably, these pathways were also changed in CVB3 and CVB3–2Am infected HeLa R19 cells with arginine, glycine, aspartate, alanine and glutamine changing in the same direction (S9A–S9D Fig). This indicates that although there is more variation in hiPSC-CMs compared to HeLa R19 cells, CVB3 and EMCV target the same pathways in both model systems.

## Discussion

We recently showed that picornaviruses remodel cellular metabolism, and predominantly nucleotide metabolism during infection [34]. It is thought that viruses actively target host metabolism to ensure the availability of sufficient building blocks for virus replication and spreading and to evade host innate immunity [18–20,22,24,26–28]. To understand whether specific viral proteins are involved in how picornaviruses remodel nucleotide metabolism, we used isotope tracing metabolomics to investigate the effect of several proteins of picornaviruses, in particular 2Apro of CVB3, L of EMCV and Lpro of FMDV on the host metabolome during infection. These proteins are known to extensively remodel the host cell during infection and suppress cellular antiviral responses. Collectively, our data suggest that 2Apro and L, but not Lpro, inhibit *de novo* nucleotide synthesis during infection. Furthermore, we show that the metabolic pathways that CVB3 and EMCV reprogram in a biologically relevant cell model, namely hiPSC-CMs, are similar to those in HeLa R19 cells.

To fully profile the impact of 2Apro of CVB3 on all nucleotide metabolic pathways, we performed $^{13}C_6$-glucose, $^{13}C_5$-glutamine, and $^{15}N_2$-glutamine isotope tracing studies during infection with wild-type CVB3 and a recombinant CVB3 virus containing a catalytically dead 2A protein (CVB3–2Am). These experiments unambiguously showed that 2Apro restricts the *de novo* nucleotide synthesis. Using another recombinant CVB3 virus, CVB3–2Bm that contains mutations in the 2B protein that slows its replication, we could show that the restriction of *de novo* nucleotide synthesis is not caused by a delay in replication but is a true functionality of 2Apro. The functionality of 2Apro is a highly conserved among the enteroviruses [11]. Therefore, it is tempting to speculate that 2Apro of other enteroviruses also inhibit *de novo* nucleotide synthesis, but further study is required to investigate this possibility. In addition to *de novo* nucleotide synthesis, the $^{15}N_2$-glutamine

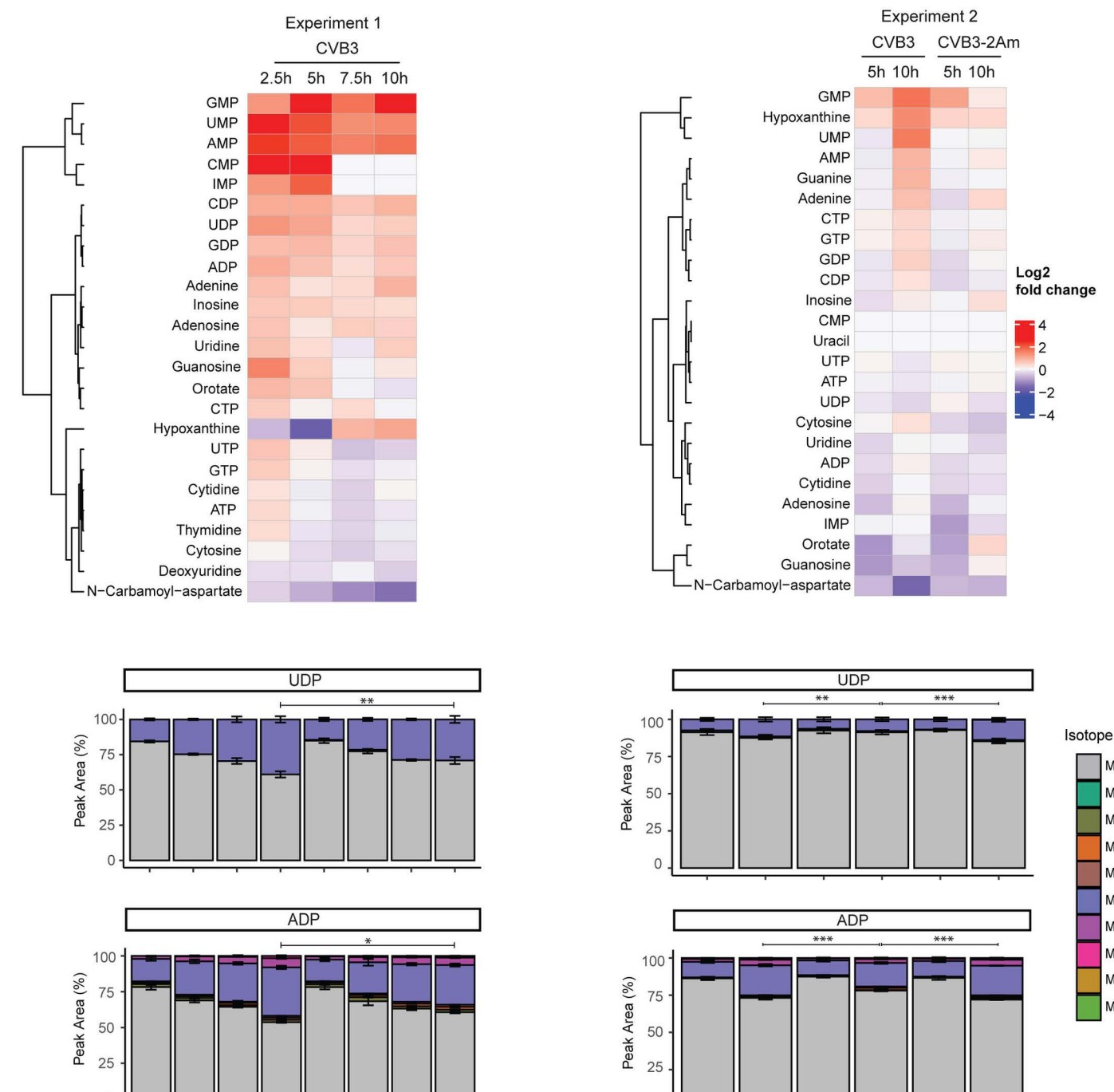

**Fig 5. CVB3 and EMCV elevate levels of purine and pyrimidine metabolites in hiPSC-CMs.** $^{13}C_6$-glucose isotope tracing study in mock-, CVB3-, CVB3-2Am or EMCV infected hiPSC-CMs (MOI 5, calculated from the HeLa R19 titer, three replicates). Cells were infected, lysed at 2.5, 5, 7.5, 10 hpi and measured by LC-MS to identify metabolites and quantify the different isotopologues. A) Heatmaps showing log2 fold changes of the purine and

pyrimidine metabolites between CVB3 infected- and mock-infected cells. Log2 fold changes are calculated based on the mean of three replicates. B) Isotopologue distribution of UDP and ADP. For statistical analysis, linear mixed effect models with an interaction of time and treatment and a random effect of replicate were performed. A rank transformation on the data was performed to ensure a normal distribution of the residuals. p-values between specific groups were calculated by performing a contrast analysis, in which the total labeled fraction was compared between groups. $*p < 0.05$, $**p < 0.01$, $***p < 0.001$.

tracing studies provided more insight into nucleotide salvage, indicating an activation of this pathway during CVB3 infection, and to a lesser extent during CVB3–2Am infection. While our $^{13}C_6$-glucose tracing studies already suggested that nucleotide salvage remained active, using $^{15}N_2$-glutamine isotope tracing we now provide evidence that the nucleotide salvage pathway is activated during CVB3 infection, rather than remaining active. Being able to differentiate between the activation or inhibition of the different arms of nucleotide metabolism is crucial, not only to understand virus replication and the host response it triggers, but also to better target these pathways using inhibitors. Numerous studies explore targeting nucleotide metabolism to inhibit virus replication [28,49,50]. A better understanding of what pathways are activated and/or inhibited during infection might lead to a more direct and targeted approach and usage of these inhibitors, and therefore ultimately a higher success rate.

Carbamoyl-phosphate synthetase 2, aspartate transcarbamylase, and dihydroorotase (CAD) is one of the two rate-limiting enzymes in the *de novo* pyrimidine synthesis. In our metabolomic screens we measure N-carbamoyl-aspartate, an intermediate in the *de novo* pyrimidine synthesis produced by CAD, and some of our screens we also measure orotate, which is produced by the second rate-limiting enzyme dihydroorotate Dehydrogenase (DHODH) directly downstream of CAD [51]. The combination of both the decreased levels of N-carbamoyl-aspartate and the decreased labeling of N-carbamoyl-aspartate and orotate during CVB3 infection suggest that there is an inhibition of CAD [34]. During CVB3–2Am infection, however, the levels of N-carbamoyl-aspartate also decreased, but the labeling of N-carbamoyl-aspartate and orotate increased compared to CVB3- and mock-infected cells. This effect is particularly visible in our $^{13}C/^{15}N$-glutamine tracing experiments, highlighting the advantage of using different labels, as both N-carbamoyl-aspartate and orotate are labeled to a greater extent by glutamine compared to glucose. The decreased levels of N-carbamoyl-aspartate combined with an increased labeling of N-carbamoyl-aspartate, orotate and the nucleotides themselves in CVB3–2Am infected cells are indicative of an increased usage of N-carbamoyl-aspartate, rather than an inhibition of its synthesis by CAD.

In the literature, there are various studies describing a possible role for CAD during infection of picornaviruses. It has been described that FMDV decreases CAD RNA and protein levels due to a decrease in HDAC1 levels, but also that depleting CAD during FMDV infection reduces viral replication [52]. In EV-A71 infected Vero cells, an upregulation of CAD expression was observed during infection and viral replication is reduced when depleting CAD [30]. During infection with other viruses, such as Hepatitis D virus (HDV), vaccinia (VACV), Kaposi's sarcoma-associated herpesvirus (KSHV) or SARS-CoV-2, CAD has been described to be activated during infection and this activation was described to be important for viral replication and NF-κB inactivation [53–56]. These studies indicate the importance of nucleotide metabolism during infection and specifically of the enzyme CAD. However, further study is warranted to fully understand the role and dynamics of CAD during picornaviral infection.

Our data indicate that 2A^pro restricts *de novo* nucleotide synthesis. 2A^pro has several main functions, including inducing NCTD and shutting off host-translation through the cleavage of eIF4G. To better understand whether one of these functionalities explains the restriction of *de novo* nucleotide synthesis by 2A^pro, we performed a comparative $^{13}C_6$-glucose isotope tracing study with CVB3, CVB3–2Am, CVB3–2Bm, CVB3–2Am + L and CVB3–2Am + L^pro. CVB3–2Am + L and CVB3–2Am + L^pro are recombinant CVB3 viruses that lack the catalytic activity of 2A^pro but include either a functional L from EMCV or a functional L^pro of FMDV. 2A^pro, L and L^pro are all proteins that are implicated in suppressing IFN responses during infection (57)]. Although these proteins share several functionalities, there are some functionalities that are unique to the different proteins. 2A^pro and L both induce NCTD, which causes the relocalization of proteins, while both 2A^pro and

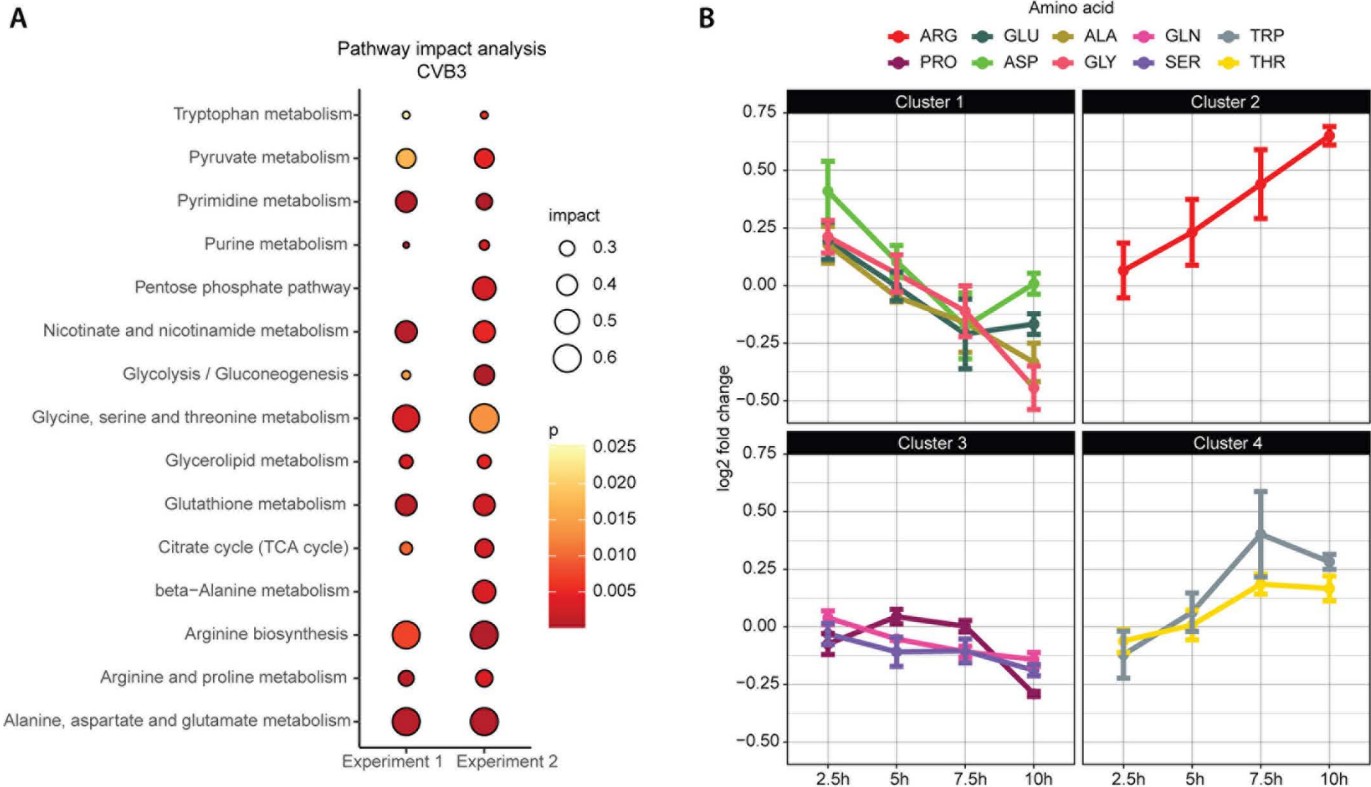

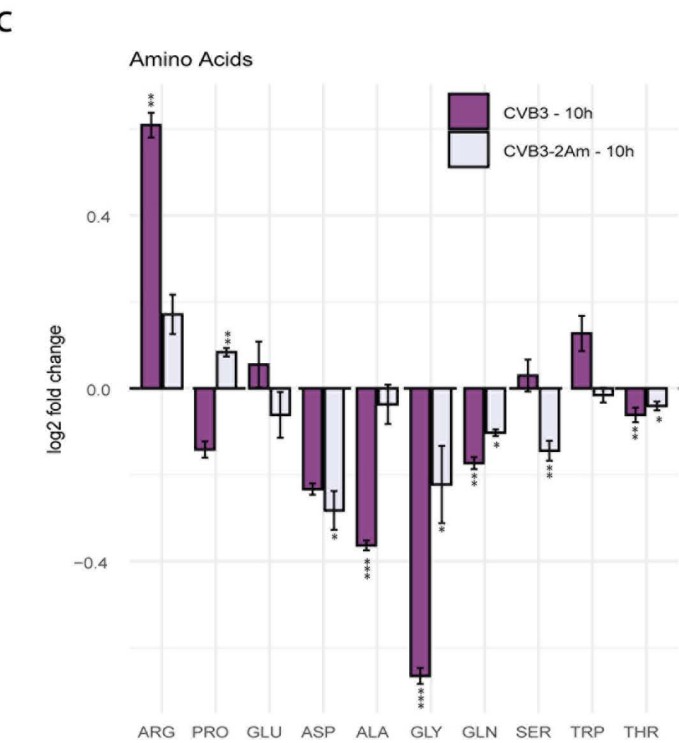

**Fig 6. CVB3 modulates amino acid metabolic pathways in hiPSC-CMs.** $^{13}C_6$-glucose isotope tracing studies in mock-, CVB3-, CVB3-2Am or EMCV-infected hiPSC-CMs (MOI 5, three replicates per experiment). Cells were infected, lysed at 2.5, 5, 7.5 or 10 hpi and measured by LC-MS to

identify metabolites. A) MetaboAnalyst pathway analysis of the two independent metabolomic experiments performed on hiPSC-CMs infected with CVB3 at 10 hpi. B) representative Log2 fold changes of amino acid levels during CVB3 infection over time (from experiment 1). C) Log2 fold changes in levels of amino acids between CVB3- or CVB3-2Am- vs mock-infected hiPSC-CMs at 10 hpi. For statistical analysis, linear mixed effect models with an interaction of time and treatment and a random effect of replicate were performed. A rank transformation on the data was performed to ensure a normal distribution of the residuals. p-values between specific groups were calculated by performing a contrast analysis, in which total peak areas were compared between groups. *p < 0.05, **p < 0.01, ***p < 0.001.

L$^{pro}$ cleave eIF4G, thereby inducing a host-translational shutoff [57]. Our $^{13}C_6$-glucose isotope tracing study with CVB3, CVB3–2Am, CVB3–2Bm, CVB3–2Am + L and CVB3–2Am + L$^{pro}$ indicates that 2A$^{pro}$ and L, but not L$^{pro}$ restrict *de novo* nucleotide synthesis. Apart from the observation that 2A$^{pro}$ and L restrict *de novo* nucleotide synthesis, the presence of 2A$^{pro}$ and L also strongly decreased N-carbamoyl-aspartate. Also, in the EMCV recombinant system, EMCV and EMCV-L$^{zn}$ + 2A$^{pro}$ reduce the levels of N-carbamoyl-aspartate the fastest. This suggests that both 2A$^{pro}$ and L inhibit CAD. However, because we could not detect orotate and used only $^{13}C_6$-glucose in these experiments, determining the precise role of CAD from these data is challenging.

Since both 2A$^{pro}$ and L restrict *de novo* nucleotide synthesis and 2A$^{pro}$ and L both cause NCTD, while L$^{pro}$ does not, it is tempting to speculate that the NCTD could be involved in limiting *de novo* synthesis. NCTD could potentially be involved in restricting *de* novo nucleotide synthesis by dislocating enzymes active in the *de novo* nucleotide synthesis pathway. Interestingly, it has been reported that *de novo* nucleotide synthesis takes place in specialized compartments in the cytosol where different enzymes of the purine or pyrimidine biosynthesis pathways are in close proximity and/or certain enzymes oligomerize based on their activation status, suggesting that localization and oligomerization are important for *de novo* nucleotide synthesis [58,59]. Additionally, it has been described that CAD is exclusively cytosolic except immediately before mitosis when the nuclear envelope breaks down [60]. The breakdown of the nuclear envelope is mimicked by the NCTD induced by 2A$^{pro}$ and it would be interesting to further study this potential change in localization of CAD. However, whether NCTD could affect the localization of the enzymes in these pathways needs to be investigated in greater detail. It has been described that 3C$^{pro}$ of CVB3 and poliovirus cleaves phosphoribosylformylglycinamidine synthase (PFAS), an enzyme in the *de novo* purine synthesis [12]. 3C$^{pro}$ is active in both CVB3 and CVB3–2Am infected cells. This indicates that although 3C$^{pro}$ might be involved in regulating the levels of *de novo* nucleotide synthesis, the effect of 2A$^{pro}$ is more substantial. The involvement of 3C$^{pro}$ exemplifies that there might be specific inhibitions of certain enzymes in the *de novo* nucleotide synthesis and that 2A$^{pro}$, L and L$^{pro}$ might not be the only proteins that affect *de novo* nucleotide synthesis.

The reason for the restriction of *de novo* nucleotide synthesis by 2A$^{pro}$ and L remains enigmatic. Considering our observations, it is possible that the restriction is secondary to the induction of NCTD, which can be accommodated by the activation of nucleotide salvage and the release of nucleotides from RNA degradation. Alternatively, it is possible that the regulation of *de novo* nucleotide synthesis during picornavirus infection is linked to innate immunity. Several enzymes in the *de novo* nucleotide synthesis have been reported to possess deamidating activity, thereby dampening the innate immune function of RIG-I, RelA or IRF3 [55,61–66]. On the other hand, depletion of nucleotides by inhibition of *de novo* nucleotide synthesis may activate innate immune responses [67,68]. Notably, inhibition of *de novo* nucleotide synthesis during CVB3 and EMCV infection does not lead to a depletion of nucleotides, due to the activation of RNA degradation and nucleotide salvage. Little is known of the consequences of *de novo* nucleotide synthesis inhibition on the activation of innate immune responses in the presence of elevated levels of RNA degradation and nucleotide salvage. Hence, the exact reason, mechanism and consequence of the restriction of *de novo* nucleotide synthesis by CVB3 and EMCV is still enigmatic and should be studied in further detail.

As HeLa R19 cells are derived from cancerous tissue and are characterized by an aberrant metabolic profile [44–47], metabolic alterations induced by picornaviruses in primary cells may differ. To study picornavirus-induced metabolic alterations in a more physiologically relevant context, we used hiPSC-CMs. Overall, our data indicate that the same metabolic

pathways are affected by CVB3 and EMCV infection in both cell types. In two independent experiments with hiPSC-CMs, we observed an increase in purine and pyrimidine metabolites as well as an increase in nucleic acid and nucleotide degradation. This implies that also in a physiologically more relevant context, nucleotide metabolism is affected by CVB3 and EMCV. However, studying *de novo* nucleotide synthesis proved to be more challenging due to the lower rate of *de novo* nucleotide synthesis in these cells.

In addition to nucleotide metabolism, alanine, arginine, proline, glutamate and aspartate metabolism was affected by CVB3 and EMCV infection in both hiPSC-CMs, as well as in HeLa R19 cells. These amino acids are part of a variety of metabolic pathways, such as nucleotide metabolism, urea cycle metabolism and protein synthesis [69–73]. Interestingly, it has been described that CVB3 increases the activity of Argininosuccinate synthase 1 (ASS1) in macrophages, an enzyme that is part of the urea cycle and consumes aspartate and citrulline to produce argininosuccinate. By activating ASS1, macrophages polarization is affected and viral myocarditis promoted [74]. Another amino acid, arginine, is known to be used to produce NO by nitric oxide synthase (NOS), but it can also be a substrate for arginase I and arginase II to produce ornithine and urea that can be used to produce polyamines. In this respect, it has been reported that NO inhibits CVB3 RNA and protein synthesis [75]. Moreover, it has been described that polyamines are beneficial for replication and translation of several viruses [76]. This is also true for picornaviruses. CVB3 binding to target cells is reduced when the cells are depleted from polyamines and $2A^{pro}$ and $3C^{pro}$ activity is reduced upon polyamine depletion [77,78]. However, when arginine is used to produce NO or polyamines a decrease in arginine levels would be expected, while we observe higher levels of arginine during CVB3 infection. For a better understanding of the observed changes, the urea cycle, polyamine and other amino acid pathways need to be studied in more detail. Together, our data underline the benefits of using multiple cellular systems to yield complementary results. Whereas the use of HeLa R19 cells allows for studying pathways that are less active in non-dividing cells, hiPSC-CMs provide biological context and reveal changes in pathways that could be overlooked when studying HeLa R19 cells only.

In conclusion, our metabolomic studies provide evidence that $2A^{pro}$ and L, but not $L^{pro}$, inhibit *de novo* nucleotide synthesis during infection. We show that CVB3 and EMCV reprogram the same pathways in HeLa R19 cells as in hiPSC-CMs. We highlight the importance of both cell types and show how they can be complementary. These insights into picornaviral modulation of cellular metabolism are important to increase our understanding of picornavirus-host interactions and to provide information on possible therapeutic targets.

## Materials and methods

### Cell lines and viruses

HeLa R19 and HEK293T cells were maintained in Dulbecco's modified Eagle's medium (DMEM; Capricorn) supplemented with 10% fetal bovine serum (FBS) and 1% Pen-Strep (Cytiva). Viral stocks were generated by propagating the virus on HeLa R19 or HEK293T cells (CVB3, CVB3–2Am, CVB3–2Am-L, CVB3–2Am-Lpro) or by transfection of run-off RNA transcripts of viral cDNA clones (CVB3–2Bm). CVB3–2Am contains the following mutations in the catalytic site of 2A: H21A, D39A, C110A and also contains a 3 CD cleavage site between P1 and P2 as previously described [10,35]. CVB3–2Bm contains mutations K[41]L and K[48]L [40]. CVB3–2Am-L and CVB3–2Am-Lpro are similar as CVB3–2Am but contain the L protein of EMCV or $L^{pro}$ of FMDV, respectively, in front of VP4 followed by a 3 CD cleavage site. Upon complete CPE, the cultures were freeze-thawed three times after which the virus was harvested and cell debris was pelleted at 4,000xg for 15 minutes. Thereafter, ultracentrifugation was used to concentrate the virus stocks (30% sucrose, 140,000 g for 16 hours, 4°C, SW32Ti rotor). Subsequently, the viruses were diluted in PBS and stored at -80. For all recombinant viruses, viral RNA was isolated (Macherey-Nagel, Nucleospin viral RNA) and sequenced to check for the correct mutations. The titers of the viral stocks were determined by end point titration on HeLa R19 cells according to the Spearman-Kärber method and expressed as 50% Tissue Culture Infectious dose (TCID50) per ml. These TCID50/ml values were then used to calculate the MOIs (TCID50/cell) used in the experiments.

## Human induced pluripotent stem cell culture and differentiation

hiPSCs were obtained from healthy donor peripheral blood mononuclear cells by Sendai virus reprogramming at Uniklinik Koln and were kindly provided by Dr. Tomo Saric [79]. The hiPSCs have previously been deposited in the European Bank for Induced Pluripotent Stem Cells (UKKi036-C, NP0143-18, EBiSC) and are registered in the online registry for human pluripotent stem cell lines (hPSCreg.eu). All experiments were conducted in accordance with the guidelines of the code of proper use of human tissue used in the Netherlands. hiPSCs were grown on a growth-factor-reduced Matrigel coating (#356231; Corning, 0.1 mg/mL) and cultured in the Essential 8 (Gibco, # A1517001) medium, which was replaced every other day. Cells were non-enzymatically passaged every 4–5 days using $0.5 \times 10^{-3}$ m EDTA (Thermo Fisher Scientific).

hiPSCs were differentiated into cardiomyocytes (CM) using a sequential Gsk3 and Wnt inhibition protocol, as previously described [80,81]. In brief, at 85% confluence, differentiation to hiPSC-CMs was initiated by changing medium to RPMI 1640 (Thermo Fisher Scientific, 11875085) with 2% B27 (B27 medium, Thermo Fisher Scientific, A1895601) and 6 µM CHIR99021 (Selleck Chemicals, S2924). Medium was changed and supplemented with 2 µM Wnt-C59 (R&D Systems, 5148) after 3 days. On day 7, the medium was replaced with RPMI 1640 with 2% B27-insulin (B27i medium, Thermo Fisher Scientific, 17504001) and on day 9 to B27i medium without glucose (Thermo Fisher Scientific, 118979020) to deplete potential non-CMs. Cells were re-plated on day 11 in B27i medium with 10% KnockOut Serum Replacement (Thermo Fisher Scientific, 108280028) and 10 µM Rock-1 inhibitor Y-27632 (Selleck Chemicals, S1049). After two days the medium was changed to B27i medium and refreshed every other day. hiPSC-CMs started beating around day 10 and were used for downstream assays from day 14, showing synchronized contractions. All cell cultures were tested negative for mycoplasma contamination using MycoAlert Kit (Lonza).

## Immunostaining hiPSCs

For each hiPSC-CM differentiation batch, hiPSC-CM differentiation purity and morphology was assessed by fluorescent immunostaining for Troponin T and Vimentin (non-CMs). Day 13 cells were prepared for immunostaining by 15 min fixation using 4% paraformaldehyde, permeabilization using 0.1% Triton-X-100 (Sigma-Aldrich) in DPBS for 10 min, and subsequent blocking with 10% normal goat serum (Sigma-Aldrich) for 30 min. The cells were then incubated at 4°C overnight with primary antibodies (Troponin T, 1:350, ab45932; Abcam and Vimentin, 1:350, V6630; Sigma) diluted in 0.1% Triton X-100 in DPBS. After DPBS washes, fluorescent labelling was performed using goat anti-rabbit Alexa fluor-488, and goat anti-mouse Alexa fluor-568 antibodies (Thermo Fisher Scientific, 1:500), and nuclear labelling using 1 µg/mL Hoechst (Thermo Fisher Scientific) for 1 hour at room temperature. After washing in DPBS and mounting in Fluoromount-G (Southern Biotech), imaging was performed using a Leica SP8X confocal microscope.

## Flow cytometry

hiPSC-CMs were detached using TrypLE select 10x (Thermo Fisher, A12177) for 10 minutes, centrifuged for 3 min at 200 x g, washed with PBS and counted. From each differentiation batch, hiPSC-CM purity was assessed using a minimum of 250.000 cells. Cells were fixed in 4% paraformaldehyde for 15 min, washed and resuspended in permeabilization buffer containing 5% BSA and 0.3% Triton X-100 in PBS for 1 hour. Cells were incubated with Troponin T - FITC (1:50, 130-119-575; Miltenyi Biotec) or isotype control antibody (REA Control Antibody (I) - FITC, human IgG1, Miltenyi Biotec) in PBS with 1% BSA and 0.06% Triton-X-100 for 30 minutes at 4°C. Cells were then washed and resuspended in flow cytometry buffer (PBS with 2% FBS + 20% EDTA) and analysed using a Cytoflex flow cytometer #A00-1–1102 (Beckman Coulter).

## Metabolite profiling and isotope tracing

HeLa R19 cells were seeded in 6 well plates at a density of 4*10^5 cells/well or hiPSC-CMs were seeded at a density of 1*10^6 in 12 well plates. The day after seeding (HeLa R19) or at the day of infection (hiPSC-CMs), the cells were infected with the

corresponding viruses (diluted in DMEM with 10% FBS unless specified otherwise) for 30 minutes at 37°C. After 30 minutes, the medium was refreshed. For HeLa R19 cells the refreshed media consisted of DMEM supplemented with 10% FBS, 2 mM glutamine ($^{15}N_2$-glutamine, $^{13}C_5$-glutamine or $^{14}N^{12}C$-glutamine) and 25 mM glucose ($^{13}C_6$-glucose (Cambridge Isotopes) or 12C-glucose). For hiPSC-CMs the refreshed media consisted of RPMI 1640 supplemented with 1% B27i and 11 mM glucose ($^{13}C_6$-glucose (Cambridge Isotopes) or 12C-glucose). At the set time points, cells were washed with ice cold PBS and lysed with 1 ml (HeLa R19 cells) or 400 µl (hiPSC-CMs) of lysis buffer (methanol/acetonitrile/H2O (2:2:1)) for metabolite extraction. Cell lysates were centrifuged at 15.000g for 15 minutes (4°C) and supernatant was collected for liquid chromatography mass spectrometry (LC-MS) analysis.

The samples were run on a Exactive mass spectrometer (Thermo Scientific) coupled with a Dionex Ultimate 3000 autosampler and pump (Thermo Scientific) or on a Q-Excative HF mass spectrometer (Thermo Scientific). Both machines operated in polarity-switching mode with spray voltages of 4.5 kV and -3.5 kV. A Sequant ZIC-pHILIC column (2.1 × 150 mm, 5 µm, guard column 2.1 × 20 mm, 5 µm; Merck) was used to separate metabolites within the samples. The solvents used are acetonitrile and eluent A (20 mM (NH4)2CO3, 0.1% NH4OH in ULC/MS grade water; Biosolve). The flow rate was set on 150 ul/min with a gradient of 20–60% of eluent A in 20 minutes. After 20 minutes, the column was washed with 80% A and re-equilibrated with 20% A.

After sample acquisition, metabolites were identified and quantified using TraceFinder software (Thermo Scientific) on the basis of the exact mass within 5 ppm as well as a set of reference standards. Peak intensities were normalized using mean peak intensities of total metabolites. Isotope distributions were corrected for natural abundance of $^{13}C$ or $^{15}N$.

### Assessing viral growth kinetics

HeLa R19 cells or hiPSC-CMs were seeded in 96 well plates with a density of 10*10^3 or 10*10^4 cells/well respectively. The next day (for HeLa R19) or at the day of infection (hiPSC-CMs), the cells were infected for 30 minutes at 37°C after which the virus was removed and replaced by DMEM+10% FBS. At the indicated time points, the plates were frozen at -80°C. After three freeze-thawing rounds, the titers of the samples were determined by end point titration on HeLa R19 according to the Spearman-Kärber method and expressed as TCID50/ml.

### Analysis & statistics

R was used to analyze the metabolomic data. Heatmaps were generated using the ComplexHeatmap R package, while PCA plots were created using the MixOmics R package [82–84]. For the cluster analysis, hclust from the stats package was used with the method 'ward.D' [85]. For the pathway impact analysis metaboanalyst.ca was used [86]. The MetaboAnalyst results were imported into R and filtered on pathways that included more than two metabolites, with a p-value of <0.05 and a pathway impact of >0.23. To statistically analyze the metabolomic data, a linear mixed effect model with an interaction of time and treatment and a random effect of replicate was performed for the depicted metabolites (using the lme4 R package [87]). Per metabolite, the normal distribution of the residuals was checked and if needed a rank transformation was performed. In case normality could still not be assumed, a non-parametric linear mixed effect model with an interaction of time and treatment and a random effect of replicate was performed (using the ARTool R package [88]). After creating the appropriate model, the model was used to perform contrast analyses to test the differences between the groups (using the emmeans R package [89]). The contrast analysis includes a Tukey test when comparing multiple treatment groups to correct for multiple testing.

### Supporting information

**S1 Fig. Glycine is marginally labeled by $^{15}N_2$-glutamine.** $^{15}N_2$-glutamine isotope tracing study in mock-, CVB3–2Am- and CVB3-infected HeLa R19 cells (MOI 5, three replicates treatment and per isotope). Cells were infected, lysed at 4,6 or 8 hpi and measured by LC-MS to identify metabolites and quantify the different isotopologues. The labeling of glycine by $^{15}N_2$-glutamine is shown.
(TIF)

**S2 Fig. Growth kinetics and characteristics of the recombinant CVB3 viruses.** A) Growth kinetics of the recombinant CVB3 viruses in HeLa R19 cells, titrated on HeLa R19 cells (mean and SD of triplicates; one experiment, MOI 5). The cells were infected, lysed at 2, 4, 6, 8 and 10 and titrated (on HeLa R19 cells) to determine the TCID50/ml.
(TIF)

**S3 Fig. 2A<sup>pro</sup> and L restrict de novo nucleotide synthesis, while L<sup>pro</sup> does not.** $^{13}C_6$-glucose isotope tracing study in mock- and CVB3-, CVB3–2Am, CVB3–2Am+L, CVB3–2Am+L<sup>pro</sup> infected HeLa R19 cells (MOI 5, three replicates). Cells were infected, lysed at 6 or 8 hpi and measured by LC-MS to identify metabolites and quantify the different isotopologues. Data obtained at 6 hpi are shown here. A) Peak areas and isotopologue distribution of UTP and GTP at 6 hpi. D) Peak areas of N-carbamoyl-aspartate at 6 hpi. For statistical analysis, linear mixed effect models with an interaction of time and treatment and a random effect of replicate were performed. A rank transformation on the data was performed to ensure a normal distribution of the residuals. For GTP, a normal distribution of the residuals could not be assumed for the fractional data and therefore a non-parametric linear mixed effect model with an interaction of time and treatment and a random effect of replicate was performed. p-values between specific groups were calculated by performing a contrast analysis, in which total labeled fraction (A, left panels) or total peak areas (A, right panels & B) were compared between groups. *$p < 0.05$, **$p < 0.01$, ***$p < 0.001$.
(TIF)

**S4 Fig. 2A<sup>pro</sup> and L restrict _de novo_ nucleotide synthesis, while L<sup>pro</sup> does not.** $^{13}C_6$-glucose isotope tracing study in mock- and EMCV-, EMCV-L<sup>zn</sup>-, EMCV-L<sup>zn</sup>+2A<sup>pro</sup>- and EMCV-L<sup>zn</sup>+L<sup>pro</sup>-infected HeLa R19 cells (MOI 5, three replicates; the mock and EMCV conditions of this experiment have already been shown in our previous work [34]). Cells were infected, lysed at 4,6 or 8 hpi and measured by LC-MS to identify metabolites and quantify the different isotopologues. two samples were removed from analysis (one replicate of 8 hpi EMCV and one replicate of 8 hpi EMCV-L<sup>zn</sup>+L<sup>pro</sup>), because of a technical defect. A) Schematic representation of EMCV and the EMCV recombinant viruses that were used in this study. B) Growth curve of EMCV and the recombinant EMCV viruses in HeLa R19 cells. C) Absolute peak areas and isotopologue distribution of UTP and ATP. D) Absolute peak areas of N-carbamoyl-aspartate. For statistical analysis, linear mixed effect models with an interaction of time and treatment and a random effect of replicate were performed. For D) A rank transformation on the data was performed to ensure a normal distribution of the residuals. For N-carbamoyl-aspartate in E), a normal distribution of the residuals could not be assumed and therefore a non-parametric linear mixed effect model with an interaction of time and treatment and a random effect of replicate was performed. p-values between specific groups were calculated by performing a contrast analysis, in which total labeled fraction (C) or total peak areas (D & E) were compared between groups. *$p < 0.05$, **$p < 0.01$, ***$p < 0.001$.
(TIF)

**S5 Fig. Growth kinetics CVB3 and EMCV in hiPSC-CMs.** A) Representative flow cytometry plot of Troponin T positive hiPSC-CMs (day 13), showing highly pure differentiations (96.5% Troponin T+). B) Representative fluorescent immunostaining of Troponin T positive hiPSC-CMs (in green) and Vimentin positive non-CMs (in red) with hoechst counterstained nuclei (in blue), showing normal hiPSC-CM sarcomere structure and high purity of differentiation. C), D) Growth kinetics of CVB3 (C) and EMCV (D) in hiPSC-CMs, titrated on HeLa R19 cells (mean and SD of triplicates; one experiment, MOIs are calculated using HeLa R19 titers). The cells were infected, lysed at 2, 4, 6, 8, 10 and 16 hpi and titrated to determine the TCID50/ml.
(TIF)

**S6 Fig. Variation in metabolic rewiring during CVB3 infection in hiPSC-CMs and compared to HeLa R19 cells.** $^{13}C_6$-glucose isotope tracing studies in mock-, CVB3- or CVB3–2Am infected hiPSC-CMs (MOI 5 TCID50/ml calculated from the HeLa R19 titer, three replicates) or HeLa R19 cells (MOI 5, three replicates). Cells were infected, lysed at 2.5,

5, 7.5, 10 hpi (hiPSC-CMs) or lysed at 2, 4, 6, 8, 10 hpi (HeLa R19 cells) and measured by LC-MS to identify metabolites and quantify the different isotopologues. A) PCA plot of the experiments with hiPSC-CMs. B) PCA plot of the experiments of both hiPSC-CMs and HeLa R19 cells.
(TIF)

**S7 Fig. CVB3 and EMCV elevate levels of purine and pyrimidine metabolites in hiPSC-CMs.** $^{13}C_6$-glucose isotope tracing study in mock-, CVB3-, CVB3–2Am or EMCV infected hiPSC-CMs (MOI 5, calculated from the HeLa R19 titer, three replicates). Cells were infected, lysed at 2.5, 5, 7.5, 10 hpi and measured by LC-MS to identify metabolites and quantify the different isotopologues. A) Heatmap showing log2 fold changes of the purine and pyrimidine metabolites between EMCV infected- and mock-infected cells. Log2 fold changes are calculated based on the mean of three replicates. B) Absolute levels of N-carbamoyl-aspartate. C) Isotopologue distribution of UDP and ADP. To statistically analyze the data, linear mixed effect models with an interaction of time and treatment and a random effect of replicate were performed. A rank transformation on the data was performed to ensure a normal distribution of the residuals. p-values between specific groups were calculated by performing a contrast analysis, in which total peak areas (B) or total labeled fraction (C) were compared between groups. *$p < 0.05$, **$p < 0.01$, ***$p < 0.001$.
(TIF)

**S8 Fig. CVB3 and EMCV infection alters amino acid levels in hiPSC-CMs.** $^{13}C_6$-glucose isotope tracing studies in mock- or EMCV-infected hiPSC-CMs (MOI 5, three replicates per experiment). Cells were infected, lysed at 2.5, 5, 7.5 or 10 hpi and measured by LC-MS to identify metabolites and quantify the different isotopologues. The isotopologue distribution is not shown in this figure. A) Log2 fold changes of amino acid levels during EMCV infection over time.
(TIF)

**S9 Fig. CVB3 modulates arginine, proline, aspartate and glutamate metabolism in HeLa R19 cells.** $^{13}C_6$-glucose isotope tracing studies in mock-, CVB3- and CVB3–2Am HeLa R19 cells (MOI 5, three replicates per experiment). Cells were infected, lysed at 2, 4, 6, 8 or 10 hpi and measured by LC-MS to identify metabolites and quantify the different isotopologues. The different isotopologues are not distinguished in this Figure. A) MetaboAnalyst pathway analysis of the two different metabolomic experiments performed on HeLa R19 cells infected with CVB3 at 8 hpi. B) MetaboAnalyst pathway analysis of the two different metabolomic experiments performed on HeLa R19 cells infected with CVB3–2Am at 8 hpi. C) Representative Log2 fold changes of amino acid levels during CVB3 infection over time (from experiment 1). D) Representative Log2 fold changes between CVB3–2Am- vs mock-infected HeLa R19 cells at 8 hpi of amino acids (from experiment 1). To statistically analyze the data, linear mixed effect models with an interaction of time and treatment and a random effect of replicate were performed. p-values between specific groups were calculated by performing a contrast analysis, in which total peak areas were compared between groups. *$p < 0.05$, **$p < 0.01$, ***$p < 0.001$.
(TIF)

**S1 Data. Used research data.**
(XLSX)

## Acknowledgments

The authors would like to gratefully acknowledge Tomo Saric (Uniklinik Koln, Germany) for providing the human iPSCs.

## Author contributions

**Conceptualization:** Lonneke V. Nouwen, Esther A Zaal, Alain van Mil, Celia R Berkers, Frank J.M. van Kuppeveld.
**Data curation:** Lonneke V. Nouwen.

**Formal analysis:** Lonneke V. Nouwen.

**Funding acquisition:** Alain van Mil, Celia R Berkers, Frank J.M. van Kuppeveld.

**Investigation:** Lonneke V. Nouwen, Inge Buitendijk, Marleen Zwaagstra, Chiara Aloise, Arno L.W. van Vliet, Jelle G Schipper, Alain van Mil.

**Methodology:** Lonneke V. Nouwen, Inge Buitendijk, Marleen Zwaagstra, Chiara Aloise, Arno L.W. van Vliet, Jelle G Schipper.

**Resources:** Alain van Mil.

**Supervision:** Esther A Zaal, Celia R Berkers, Frank J.M. van Kuppeveld.

**Visualization:** Lonneke V. Nouwen, Alain van Mil.

**Writing – original draft:** Lonneke V. Nouwen.

**Writing – review & editing:** Alain van Mil, Esther A Zaal, Celia R Berkers, Frank J.M. van Kuppeveld.

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
