## [Decision Letter · Decision Letter 0]

PPATHOGENS-D-25-00688

Identification of picornavirus proteins that inhibit de novo nucleotide synthesis during infection

PLOS Pathogens

Dear Dr. van Kuppeveld,

Thank you for submitting your manuscript to PLOS Pathogens. After careful consideration, we feel that it has merit but does not fully meet PLOS Pathogens's publication criteria as it currently stands. Therefore, we invite you to submit a revised version of the manuscript that addresses the points raised during the review process.

Please submit your revised manuscript within 30 days Jun 23 2025 11:59PM. If you will need more time than this to complete your revisions, please reply to this message or contact the journal office at plospathogens@plos.org. Please include the following items when submitting your revised manuscript:

We look forward to receiving your revised manuscript.

Kind regards,

Peter Sarnow

Academic Editor

PLOS Pathogens

Sonja Best

Section Editor

PLOS Pathogens

Sumita Bhaduri-McIntosh

Editor-in-Chief

PLOS Pathogens

orcid.org/0000-0003-2946-9497

Michael Malim

Editor-in-Chief

PLOS Pathogens

orcid.org/0000-0002-7699-2064

**Journal Requirements:**

At this stage, the following Authors/Authors require contributions: Lonneke Nouwen, Esther A Zaal, Inge Buitendijk, Marleen Zwaagstra, Chiara Aloise, Arno L.W. van Vliet, Jelle G Schipper, Alain Van Mil, Celia R Berkers, and Frank J.M. van Kuppeveld. Please ensure that the full contributions of each author are acknowledged in the "Add/Edit/Remove Authors" section of our submission form.

- TM on page: 24.

4) Your manuscript's sections are not in the correct order.  Please amend to the following order: Abstract, Introduction, Results, Discussion, and Methods

5) Please upload all main figures as separate Figure files in .tif or .eps format. For more information about how to convert and format your figure files please see our guidelines: 

6) We have noticed that you have uploaded Supporting Information files, but you have not included a list of legends. Please add a full list of legends for your Supporting Information files after the references list.

7) Some material included in your submission may be copyrighted. According to PLOSu2019s copyright policy, authors who use figures or other material (e.g., graphics, clipart, maps) from another author or copyright holder must demonstrate or obtain permission to publish this material under the Creative Commons Attribution 4.0 International (CC BY 4.0) License used by PLOS journals. Please closely review the details of PLOSu2019s copyright requirements here: PLOS Licenses and Copyright. If you need to request permissions from a copyright holder, you may use PLOS's Copyright Content Permission form.

Potential Copyright Issues:

i) Figure 4B. Please confirm whether you drew the images / clip-art within the figure panels by hand. If you did not draw the images, please provide (a) a link to the source of the images or icons and their license / terms of use; or (b) written permission from the copyright holder to publish the images or icons under our CC BY 4.0 license. Alternatively, you may replace the images with open source alternatives. See these open source resources you may use to replace images / clip-art:

8) We note that your Data Availability Statement is currently as follows: "All relevant data are within the manuscript and its Supporting Information files.". Please confirm at this time whether or not your submission contains all raw data required to replicate the results of your study. Authors must share the “minimal data set” for their submission. PLOS defines the minimal data set to consist of the data required to replicate all study findings reported in the article, as well as related metadata and methods (https://journals.plos.org/plosone/s/data-availability#loc-minimal-data-set-definition).

**Reviewers' Comments:**

Reviewer's Responses to Questions

**Part I - Summary**

Reviewer #1: Nouwen and colleagues present a comprehensive investigation of metabolic changes in cells infected with picornaviruses focusing on the nucleotide and amino acid synthesis pathways. Picornaviruses are known to induce rapid and profound changes in cellular metabolism, but the mechanism and significance of many of such infection-induced perturbations remain poorly understood. Using metabolic tracing with different labeled compounds which allowed complementary assessment of diverse metabolic pathways, the authors show that infection with Coxsackievirus B3, an enterovirus, and encephalomyocarditis virus, a cardiovirus, inhibits de novo nucleotide synthesis but activates the nucleotide salvage pathways. Using elegant replication systems where they could inactivate and substitute specific viral proteins they convincingly demonstrate that enterovirus protease 2A and cardiovirus leader protein L are largely responsible for the observed changes in nucleotide metabolism. They further speculate that given that both proteins induce inhibition of nucleocytoplasmic trafficking, albeit by different mechanisms, the latter may be required for the inhibition of de novo nucleotide synthesis. The major findings obtained in HeLa cells were also corroborated in a more physiological system of cardiomyocytes differentiated from pluripotent cells.

The experiments are performed on an excellent technical level and will represent an important resource for everyone interested in metabolic changes in infected cells.

Reviewer #2: Building upon their previous report (citation #34; PLos Path 2024), Nouwen and colleagues report that the 2A protease of CVB3 and the leader protein of EMCV inhibit de novo nucleotide synthesis. These conclusions are substantiated by experiments exploiting state-of-the-art isotope tracing metabolomics and a clever panel of genetically engineered coxsackieviruses (Fig. 4B): CVB3, CVB3 with a debilitating 2A protease mutation, and derivatives thereof expressing the EMCV leader protein or the FMDV leader protease. As the authors highlight (Fig. 4A), the EMCV and FMDV leader proteins share some phenotypic traits with CVB 2A protease, allowing the authors to test whether previously established phenotypes of CVB3 2A protease (nucleocytoplasmic trafficking disorder, cap-dependent host translation shut-off via eIF4G cleavage or IFN suppression) influence de novo nucleotide synthesis. Intriguingly, CVB3 2A protease and EMCV leader protein inhibit de novo nucleotide synthesis, while FMDV leader protease does not. The inhibition of de novo nucleotide synthesis was clear in CVB3-infected HeLa R19 cells (where de novo nucleotide synthesis is a normal aspect of cellular metabolism). However, similar phenotypes were not completely evident in human induced pluripotent stem cell-derived cardiomyocytes (where a lower rate of de novo nucleotide synthesis confounds some comparisons with phenotypes in HeLa cells).

The authors conclude that the CVB3 2A protease and the EMCV leader protein, but not the FMDV leader protease, inhibit de novo nucleotide synthesis. Furthermore, they conclude that the nucleotide salvage pathway is activated during CVB3 infection, providing a source of nucleotides for virus replication. The authors cautiously suggest that nuclear pores and/or nucleocytoplasmic trafficking could be linked to de novo nucleotide synthesis, and modifications of nuclear pores by CVB3 2A protease and the EMCV leader protein could disrupt and/or relocalize nuclear pore adjacent enzymes involved in de novo nucleotide synthesis.

Critique: This is a novel investigation with important insights into nucleotide metabolism during enterovirus infections. It appears that the nucleotide salvage pathway, rather than de novo nucleotide synthesis, is activated during CVB3 infection, providing a source of nucleotides for virus replication. The authors provide strong evidence to show that the CVB3 2A protease and the EMCV leader protein, but not the FMDV leader protease, inhibit de novo nucleotide synthesis. However, it is unclear whether the inhibition of de novo nucleotide synthesis by viral proteins benefits virus replication, or is otherwise biologically relevant in the context of enterovirus infections in vitro or in vivo. Is the inhibition of de novo nucleotide synthesis biologically relevant, or an incidental phenotype?

Reviewer #3: This manuscript focuses on the nucleotide metabolism control during picornavirus infection. Specifically, the authors used isotope labeling techniques in metabolomics to intricately trace the pathways of nucleotide metabolism to identify how a specific viral protein 2A in Cocksackie B3 virus, controlled these pathways. They discovered that the wild type virus down regulated de novo biosynthesis pathways and this down regulation was dependent upon 2A activity. Additionally, they show that this phenotype is reproducible in additionaly cell types and across picornaviruses.

Isotope tracing is extremely challenging to do, and the authors have carried out well-controlled, systematic analyses that are very in-depth and elegantly designed. The results are very clear and informative for the field and I see no problems. Issues below are to improve clarity.

minor issues:

In the discussion, it would be good to hear hypotheses about why these viruses might prefer to down regulate de novo synthesis and instead rely on salvage pathways of nucleotide metabolism.

CAD is not defined.

line 508, sentence is not complete.

**Part II – Major Issues: Key Experiments Required for Acceptance**

Reviewer #1: One missing point that should be addressed, at least in the Discussion, is the significance of the metabolic changes observed for the viral infection. Given that the viruses defective in the inhibition of de novo nucleotide synthesis can replicate reasonably well, and that the magnitude of the effect is different in different cell types, do the authors consider the variations in nucleotide metabolism to be essential for the development of infection, especially in the in vivo conditions? Can the inhibition of the nucleotide salvage pathway on the development of infection be tested?

Reviewer #2: 1. Biological significance. Is the inhibition of de novo nucleotide synthesis advantageous for enteroviruses, or an incidental phenotype? It is unclear whether the inhibition of de novo nucleotide synthesis by viral proteins benefits virus replication, or is otherwise biologically relevant in the context of enterovirus infections in vitro or in vivo.

Reviewer #3: None

**Part III – Minor Issues: Editorial and Data Presentation Modifications**

Reviewer #1: Line 188 of, not off

Line 233. The expression “favorable function upon expression” is vague and unnecessary. It is sufficient to state that mutations in 2B result in the replication delay.

Reviewer #2: 1. Line 151 and elsewhere throughout the manuscript. Clarify the MOI used for experimental infections. Virus titers were obtained by limiting dilution (TCID50 per ml); Does 5 TCID50 per ml = 5 TCID50 per cell for experimental infections?

2. Line 177. ...reduced or deduced?

3. Theoretical consideration. Are there any limiting metabolites evident at any point during infection? Are cells metabolically exhausted at any point during infection, where low amounts of one or another metabolite could limit viral gene expression and replication?

4. Mechanistic Gaps. It remains unclear how CVB3 2A protease and the EMCV leader protein inhibit de novo nucleotide synthesis. The discussion provided some very nice speculation (for future studies) to address mechanistic gaps.

Reviewer #3: minor issues:

In the discussion, it would be good to hear hypotheses about why these viruses might prefer to down regulate de novo synthesis and instead rely on salvage pathways of nucleotide metabolism.

CAD is not defined.

line 508, sentence is not complete.

PLOS authors have the option to publish the peer review history of their article (what does this mean? ). If published, this will include your full peer review and any attached files.

**Do you want your identity to be public for this peer review?** For information about this choice, including consent withdrawal, please see our Privacy Policy .

Reviewer #1: No

Reviewer #2: No

Reviewer #3: No

**Figure resubmission:**
---

## [Editor Report · Decision Letter 1]

Dear Prof. Dr. van Kuppeveld,

We are pleased to inform you that your manuscript 'Identification of picornavirus proteins that inhibit de novo nucleotide synthesis during infection' has been provisionally accepted for publication in PLOS Pathogens.

Best regards,

Peter Sarnow

Academic Editor

PLOS Pathogens

Sonja Best

Section Editor

PLOS Pathogens

Sumita Bhaduri-McIntosh

Editor-in-Chief

PLOS Pathogens

orcid.org/0000-0003-2946-9497

Michael Malim

Editor-in-Chief

PLOS Pathogens

orcid.org/0000-0002-7699-2064
---

## [Editor Report · Acceptance letter]

Dear Prof. Dr. van Kuppeveld,

We are delighted to inform you that your manuscript, "Identification of picornavirus proteins that inhibit de novo nucleotide synthesis during infection," has been formally accepted for publication in PLOS Pathogens.

Best regards,

Sumita Bhaduri-McIntosh

Editor-in-Chief

PLOS Pathogens

orcid.org/0000-0003-2946-9497

Michael Malim

Editor-in-Chief

PLOS Pathogens

orcid.org/0000-0002-7699-2064